Controller-driven vector autoregression model for predicting content popularity in programmable named data networking devices

Qaiser Firdous 1
Hussain Mudassar 2
http://orcid.org/0000-0002-3914-2503 Ahad Abdul 2 3 4 ahad9388@gmail.com
http://orcid.org/0000-0002-3394-6762 Pires Ivan Miguel 5 impires@ua.pt
1 Department of Computer Science, University of Sialkot , Sialkot , Pakistan
2 Knowledge Unit of Systems and Technology, University of Management and Technology , Sialkot , Pakistan
3 School of Software, Northwestern Polytechnical University , Xian, Shaanxi , China
4 Department of Electronics and Communication Engineering, Istanbul Technical University (ITU) , Istanbul , Turkey
5 Instituto de Telecomunicações, Escola Superior de Tecnologia e Gestão de Águeda, Universidade de Aveiro , Águeda , Portugal
Sajid Ullah Syed
Electronic publication date: 2024 Feb 8
Publication date: 2024
Volume: 10
Electronic Location ID: e1854
Received 2023 Nov 20; Accepted 2024 Jan 11
Copyright: © 2024 Qaiser et al.
Copyright year: 2024
Copyright holder: Qaiser et al.
License: This is an open access article distributed under the terms of the Creative Commons Attribution License, which permits unrestricted use, distribution, reproduction and adaptation in any medium and for any purpose provided that it is properly attributed. For attribution, the original author(s), title, publication source (PeerJ Computer Science) and either DOI or URL of the article must be cited.
License URL: https://creativecommons.org/licenses/by/4.0/

Keywords: Information centric networking (ICN), NDN, SDN, Content caching, Caching placement, Vector autoregression (VAR)

Funding: FCT/MEC national funds The FEDER-PT2020 Partnership Agreement UIDB/50008/2020 This work is funded by FCT/MEC through national funds and co-funded by the FEDER-PT2020 partnership agreement under the project UIDB/50008/2020. The funders had no role in study design, data collection and analysis, decision to publish, or preparation of the manuscript.

==============================
Named Data Networking (NDN) has emerged as a promising network architecture for content delivery in edge infrastructures, primarily due to its name-based routing and integrated in-network caching. Despite these advantages, sub-optimal performance often results from the decentralized decision-making processes of caching devices. This article introduces a paradigm shift by implementing a Software Defined Networking (SDN) controller to optimize the placement of highly popular content in NDN nodes. The optimization process considers critical networking factors, including network congestion, security, topology modification, and flowrules alterations, which are essential for shaping content caching strategies. The article presents a novel content caching framework, Popularity-aware Caching in Popular Programmable NDN nodes (PaCPn). Employing a multi-variant vector autoregression (VAR) model driven by an SDN controller, PaCPn periodically updates content popularity based on time-series data, including ‘request rates’ and ‘past popularity’. It also introduces a controller-driven heuristic algorithm that evaluates the proximity of caching points to consumers, considering factors such as ‘distance cost,’ ‘delivery time,’ and the specific ‘status of the requested content’. PaCPn utilizes customized DATA named packets to ensure the source stores content with a valid residual freshness period while preventing intermediate nodes from caching it. The experimental results demonstrate significant improvements achieved by the proposed technique PaCPn compared to existing schemes. Specifically, the technique enhances cache hit rates by 20% across various metrics, including cache size, Zipf parameter, and exchanged traffic within edge infrastructure. Moreover, it reduces content retrieval delays by 28%, considering metrics such as cache capacity, the number of consumers, and network throughput. This research advances NDN content caching and offers potential optimizations for edge infrastructures.

Introduction

The Internet’s architecture has revolved around Internet Protocol (IP)-based communication for several decades. Based on a host-centric model, IP-based networks face challenges in efficiently distributing content across numerous autonomous systems. This distribution process introduces various overheads, including content acquisition, content management, and security, making IP networks less suitable for content-based applications such as video sharing, e-commerce, and social media networking (Xylomenos et al., 2013). This shift has led to the dominant use of the Internet as a content distribution network.

Two prominent technologies have been proposed for efficiently disseminating content across networks: peer-to-peer (P2P) and content delivery network (CDN). In P2P networks, all participating peers can act as clients and servers, with the added functionality of caching shared content throughout the network infrastructure (Magharei et al., 2013). Unfortunately, P2P systems have encountered challenges related to inter-ISP traffic, leading to suboptimal performance in terms of content availability. In contrast, CDN networks utilize central servers to replicate content across geographically distributed servers, ensuring rapid retrieval. However, CDNs have also faced performance challenges due to the traffic engineering practices of Internet Service Providers (ISPs).

To address these shortcomings and facilitate content distribution, the research community has introduced a novel and clean-slate architecture known as information centric network (ICN). Information-centric networking (ICN) is a communication paradigm that prioritizes content over physical location, enabling in-network caching where information objects are named independently of their locations (Zhang, Luo & Zhang, 2015). Instead of relying on IP addresses, users access information objects through their names, and various segments of these objects are cached across network devices like switches and routers. This fine-grained caching optimally utilizes network buffers, creating multiple pathways for on-demand content delivery by caching individual chunks or packets within an information object.

ICN offers simplified content access, improved distribution, enhanced security, and efficient in-network caching, exemplified by caching architectures such as Data-Oriented Networks (DONA) and Named Data Networking (NDN). In DONA, domains form provider/customer/peer relationships using resolution handlers (RH) for managing content registration tables (Vasilakos et al., 2015). In contrast, the NDN architecture establishes consumer/provider/producer relationships, enabling all NDN nodes, including switches, routers, and gateways, to cache substantial content aligned with user interests (Tavasoli, Saidi & Ghiasian, 2022). NDN communication involves INTEREST packets initiated by users to request content and DATA packets delivered by the network in response. These DATA packets use a hierarchical, human-readable naming scheme for uniqueness and security (Liu et al., 2017a).

The NDN delivery network comprises three fundamental components: consumers, providers, and producers (Kalghoum, Gammar & Saidane, 2018). (i) Consumers: Users who send an INTEREST packet into the network to acquire the desired data.

(ii) Providers: NDN nodes that cache incoming DATA packets in the path between the content requestor and creator, containing three functional components: content store, pending interest table, and forwarding information base. • Content store (CS): The node’s cache memory stores actual content from DATA packets, temporarily maintaining information in a table with headings such as ‘Content names’ and ‘Actual data.’

• Pending interest table (PIT): Records entries for all incoming INTEREST packets along with their requested interfaces, providing details about the interface from which incoming packets arrived. Information such as ‘Content prefix-name’ and ‘Incoming interface’ is recorded in the PIT table.

• Forwarding information base (FIB): Maintains forwarding rules to determine the next route for packet forwarding. It provides information about which NDN nodes in the network cache the requested packet, recording ‘Content prefix-name’ with their provider, referred to as ‘Next-hop,’ in the FIB table.

(iii) Producers: Content creators who maintain records of all DATA packets, documenting all content-related information.

Software defined networking and information-centric networking

Software-defined networking (SDN) is an emerging network architecture that separates the control plane from the data-forwarding plane, enabling flexible control over network infrastructure (Mughees et al., 2023). Two main components play crucial roles in SDN-based content delivery networks, such as information-centric networking (ICN). The central entity, known as the SDN controller, manages and maintains the network topology to ensure the availability of various named contents received by participating CR routers. The second component involves a straightforward forwarding approach, with caching devices responsible for routing the requested packet along the path directed by the SDN controller (Ahad et al., 2017).

The integration of ICN into the SDN framework yields numerous advantages (Zhang et al., 2019; Mahmood et al., 2018). SDN’s intrinsic flexibility facilitates real-time dynamic routing adjustments, optimizing content delivery paths. It, coupled with centralized control, enhances content placement, effectively reducing latency and improving the overall user experience. Rigorous security measures, encompassing robust access control, encryption, and threat detection, are systematically implemented, ensuring a comprehensive and steadfast security stance.

Moreover, scalability enhancements have been introduced to cater to the growing demands on the network. Integrating emerging protocols and technologies ensures ongoing compatibility with evolving standards and innovative communication methods, highlighting the adaptability of the NDN framework. A pivotal benefit is evident in SDN-controller-aware caching within the NDN delivery system, leading to improved network routing and caching strategies (Mateen et al., 2023). (i) Routing: In SDN-based NDN delivery systems, rules, and routing decisions are vital for efficient data delivery. Centralized control is achieved by centralizing network management. When a user sends an INTEREST packet, the SDN controller programs the forwarding information base (FIB) table with routing rules derived from content names (Kalafatidis et al., 2022). These rules are strategically placed in the FIB table to map each content name to a specific route or set of routes. When an INTEREST packet is sent, content routers (CR) consult the FIB table to determine the best route for fulfilling the request.

(ii) Caching: Caching in SDN-based NDN systems enhances data retrieval efficiency by strategically placing caches at network points like routers and switches. These caches store frequently accessed data, enabling swift data delivery upon request. Unlike traditional IP-based networks, SDN-based content-delivery networks cache data based on its unique name (Aldaoud et al., 2023). Its distributed caching approach reduces data duplication, minimizes network congestion, and fosters faster content delivery, ultimately enhancing overall network performance and user experience.

This article delves into popularity-based content placement in programmable NDN devices, employing a predictive time-series VAR model to anticipate content popularity trends. Concurrently, a heuristic algorithm, driven by an SDN controller and based on topological features, is proposed to identify popular placements for NDN devices. Additionally, this article aims to reduce miss rates by evaluating the local cache freshness of each received DATA packet, thereby avoiding the prefetching of frequently accessed content from external sources. Furthermore, to address data redundancy, a system of custom-named packets is implemented to prevent intermediate nodes from caching forwarded packets.

Literature review

In recent years, several caching schemes have been introduced to enhance the performance of content delivery networks (CDNs). These schemes can be broadly categorized into two primary classes: decentralized and centralized caching schemes.

De-centralized caching schemes

Decentralized caching schemes have become prominent. Cache Everything Everywhere (CEE), which involves caching content at each router, increases redundancy while decreasing cache diversity (Chai et al., 2012). CacheFilter addresses redundancy, using a FlagBit to decide whether to cache or forward content (Feng et al., 2015). WAVE, a chunk-based caching scheme based on the content frequency and inter-chunk distance, was introduced by Cho et al. (2012). Probability Caching (PopCache) is a probability-driven scheme that caches content near consumers with high probability values and at a distance from low probability ones (Suksomboon et al., 2013). Furthermore, Random Caching (RC) utilizes 0 to 1 probability values for caching (Tarnoi et al., 2014).

Recent research has highlighted popularity-driven caching strategies. The push-down push-up (PDPU) policy dynamically moves content closer to users as its popularity increases, selecting cached content based on popularity (Nour et al., 2020). Another approach coordinates and shares popularity information among network routers, forming a popularity-aware caching system (Li et al., 2012). As presented in a different study, Most Popular Content (MPC) allows routers to cache content once it reaches a locally preset popularity threshold, with each router individually setting its threshold (Bernardini, Silverston & Festor, 2013). In popular content caching, Dynamic Fine-Grained Popularity-Based Computing (D-FGPC) is introduced as an extension of MPC. D-FGPC employs a flexible popularity threshold based on Interest frequency and cache capacity (Ong et al., 2014). Another strategy in the context of popularity-driven placement entails the implementation of a time-series autoregressive (AR) model (Liu et al., 2019). This approach enriches our understanding of content popularity dynamics by deliberately considering lagged orders associated with relevant variables.

As exemplified by CRCache and BEACON, Centrality router-based optimal placement schemes strategically leverage content popularity and key routers to optimize content placement for efficient delivery (Wang et al., 2014; Xiaoqiang, Min & Muqing, 2016). As demonstrated in Zheng et al. (2019), collaborative caching schemes consider content popularity and router betweenness centrality. It involves storing popular content on highly central routers and less popular content on routers with lower centrality. Another placement scheme is introduced, making popularity and placement decisions based on the exponential weighted moving average (EWMA) and the mathematical derivation of node centrality (Liu et al., 2021). Despite its effectiveness, the method encounters delays associated with the cache’s high proximity to users.

To address the challenge, Popularity-Aware Caching (PaCC) enhances content proximity to consumers by incorporating hop-distance awareness in placement decisions (Amadeo et al., 2022). However, hop-distance-based placement entails inefficiencies for large networks. Another method presented in Dutta et al. (2022) makes caching decisions based on node distance cost and congestion cost to optimize content placement decisions. Nevertheless, it suffers from suboptimal performance in retrieval delays due to its high computational cost. Subsequently, the Caching Popular Fresh Content (CPFC) strategy is introduced to reduce miss rates and retrieval delays, dynamically assessing both popularity and content freshness for cache suitability (Amadeo et al., 2020). In the context of caching policies for edge and core routers, as discussed in Amadeo et al. (2021), the approach prioritizes placing the most popular content at the edge and popular, long-lasting content on core routers.

The existing research on decentralized-based cache placement schemes has made valuable contributions, but has certain limitations and gaps. Some schemes utilize probabilistic caching based on content request rates, potentially resulting in suboptimal cache hit rates due to fixed probability thresholds. Others rely on popularity-driven metrics, often leading to reduced cache hit rates by focusing on a single factor, such as frequency rate. Some schemes incorporate betweenness centrality for cache placement, but it may cause delays in content retrieval, especially when the content provider is distant from consumers. Additionally, specific caching criteria consider popularity and non-transient freshness. However, no mechanism exists to evaluate each cached content’s local residual freshness period, leading to cache miss rates. Table 1 summarizes the objectives and limitations of the different cache placement schemes.

Table 1 Summary of existing cache placement schemes.

Ref	Cache placement method	Objectives	Limitations	
Chai et al. (2012)	All the intermediate nodes cache content	Align demanded content with requesting users	Data duplication, limited content diversity	
Cho et al. (2012)	Cache chunks by frequency	Stored chunks by related distance	Single-factor popularity prediction reduces cache hit ratio	
Suksomboon et al. (2013)	Cache by probability metric	Assign [0,1] probability to all named packets	Static threshold caused lower hit rate	
Nour et al. (2020)	Place content by request count	Get popular content closer to users	Greater distance results in more hops	
Bernardini, Silverston & Festor (2013)	Cache data with popularity metric	Establish and maintain local popularity tables within nodes	Single-factor estimates achieved suboptimal cache performance	
Ong et al. (2014)	Cache popular content with flexible threshold	Identify popular placements through topology features	Delays due to router’s low user proximity	
Wang et al. (2014)	Cache content based on popularity-routers correlation	Identify important routers based on their distribution power	Placement in centrality nodes increases hop-count	
Xiaoqiang, Min & Muqing (2016)	Central nodes caching based on discrete request rates	Select contents based on their discrete arrival rates	Ignoring historical data led to low cache hit rate	
Amadeo et al. (2022)	Caching on closeness-aware nodes	Create hop-count based closeness metric for content providers and consumers	Computationally expensive due to updates at forwarding nodes	
Amadeo et al. (2020)	Cache content by popularity and freshness	Base caching decisions on a combination of popularity and freshness	High cache miss rate due to fixed freshness threshold	
Liu et al. (2017b)	Popularity prediction in SDN delivery network using auto-encoders	Forecast popular content from request arrivals	No method defined to select optimal placements	
Asmat et al. (2020)	Cache frequently accessed content with in-band communication	Nodes keep local and network popularity tables	The computational overhead increases due to dual popularity table functionality.	
Zha et al. (2022)	Dynamic threshold for popular content caching	Select content using exponential weighted average	Limited history-based selection led to low cache hit	
Liu et al. (2019)	Autoregressive model-driven popular content placement in NDN devices	Maintained the lagged values of relevant popularity variables	High retrieval delays due to absence of strategical placement scheme.	

Centralized caching schemes

Researchers propose a central-entity-based caching approach within Information-Centric Networks (ICN) to optimize resource management. An autoencoder-based model predicts content popularity by analyzing spatial-temporal data collected by the SDN controller. This model empowers routers to employ a softmax classifier, selecting crucial nodes based on betweenness centrality to strategically place popular content (Liu et al., 2017b; Mughees et al., 2021). In the article (Narayanan et al., 2018), the authors employed the Long Short-Term Memory (LSTM) model for popularity prediction, but the lack of a strategically implemented placement approach results in heightened retrieval latency.

In recent work (Dudeja et al., 2022; Liu et al., 2022), an SDN-based secure content caching and forwarding approach was introduced. Routers maintain an index table in their cache memory with unique signatures for data. However, it is noteworthy that the implemented placement strategy for content caching is based on frequency count, which may not be sufficient to capture the changing trends over time, resulting in heightened delays in content retrieval.

While centralized-based caching in networks has utilized topological characteristics like betweenness centrality for content placement, this can result in longer content delivery times due to central nodes being distant from consumers. Recent research introduced closeness-aware metrics based on hop counts, which may increase computational and delivery costs without accounting for variable link capacities and latency. Some studies implement popularity-aware caching across multiple autonomous systems, basing caching decisions on request counts rather than consumer historical requests, potentially leading to suboptimal cache hit rates.

The limitations identified in the current body of research underscore the necessity for our study, aiming to bridge these gaps and present a comprehensive solution by introducing a novel caching policy. This research advocates for centralized popularity-aware placement in popular NDN nodes (PaCPn), where content popularity is calculated not solely based on request counts, as seen in existing studies, but also considers previous lag values for each requested packet to predict future popular content. Consequently, a multi-variant Vector Autoregression (VAR) popularity model is proposed, anticipating future popular content based on historical values of the frequency count and previous popularity scores.

To improve the efficiency of content delivery and minimize delivery times, we introduce a closeness-aware heuristic algorithm for caching popular content. This algorithm assesses node proximity through a metric divided into distance cost to consumers and content delivery time. Diverging from studies that rely on hop distance-based closeness, we leverage a content frequency rate to ascertain a node’s proximity to specific content. The preferred caching location is identified as a node with a high closeness metric for a given content.

Problem statement

Caching in named data networking (NDN) enhances content delivery efficiency. NDN replaces traditional IP-based networks with a data-centric approach, where content is named and cached at intermediate nodes. Popularity-based caching placement strategies gain importance in NDN to optimize content access. By prioritizing frequently requested content for caching, these strategies use the intrinsic data-centric nature of NDN, ensuring that popular content is readily available at closer network locations. It minimizes retrieval delays while enhancing the cache hit rate.

In network environments, content popularity optimization is crucial for efficient caching strategies, significantly impacting both performance and resource utilization. “Performance” encompasses the overall efficiency, speed, and responsiveness of the caching system, achieved through strategically identifying and caching frequently requested content. It minimizes content retrieval latency, ensuring the swift delivery of popular data from closer network locations and contributing to a seamless user experience.

Moreover, content popularity optimization is crucial for optimizing “resource utilization.” By focusing on caching frequently requested content, redundant data retrieval requests are minimized, conserving bandwidth and reducing server load. This efficient resource allocation ensures that network resources are used effectively, fostering a cost-effective and sustainable caching infrastructure. In essence, content popularity optimization enhances both content delivery performance and the judicious utilization of network resources.

NDN, inherently distributed, introduces complexities in caching and forwarding decisions due to its decentralized nature, resulting in inefficiencies such as suboptimal content retrieval and resource allocation. The inefficiency arises from the lack of centralized control, making coordinating caching strategies effectively across the network challenging. Recognizing these challenges, adopting a centralized Software-defined networking (SDN) environment becomes imperative. The deployment of content popularity placement schemes within a centralized SDN proves pivotal in addressing these inefficiencies. The SDN controller strategically caches frequently requested content, effectively mitigating content retrieval latency and optimizing resource allocation, enhancing overall network performance and significantly improving data availability. With a higher percentage of cached content, SDN ensures a more efficient and responsive network, providing a comprehensive solution to improve efficient content delivery and data availability within the network.

The accompanying Fig. 1 illustrates a network diagram of the default Cache Everything Everywhere (CEE) placement scheme, where NDN nodes cache the packets passing through them. In this figure, the nodes cache the prefixes referred to as ‘/prefix/content1’ and ‘/prefix/content5’. All the content routers (CRs) on the left cache the ‘/prefix/content1’ name prefix, while nodes on the right store the ‘/prefix/content5’ name prefix. This caching behavior occurs during communication between CR2 and the content producer and between CR3 and the content producer. By default, the NDN network implements the CEE caching scheme with least recently used (LRU)–based replacement, leaving copies of data at each NDN node. This content replication at every NDN node can lead to data redundancy problems. Nevertheless, previous caching schemes, such as Leave Copy Down (LCD) and Move Copy Down (MCD), have been introduced to mitigate data duplication.

Figure 1 NDN network with default CEE caching scheme.

Specific caching policies ground their decisions on content popularity to improve cache diversity. However, the rigidity of fixed popularity threshold values may lead to an escalation in inter-exchange traffic (Suksomboon et al., 2013; Bernardini, Silverston & Festor, 2013; Nguyen et al., 2019). Subsequently, specific popularity-aware caching approaches incorporate popularity and freshness metrics as decision factors (Ong et al., 2014; Dudeja et al., 2022; Liu et al., 2022; Gupta et al., 2023). Unfortunately, they achieve suboptimal performance in maximizing cache hit rates because their caching decisions are primarily based on request rates, regardless of the network’s topological characteristics. In contrast, existing work employs betweenness centrality-based cache placement schemes to optimize the cache hit rate (Wang et al., 2014; Xiaoqiang, Min & Muqing, 2016; Zheng et al., 2019). These schemes base their caching decisions on the popularity metric and a set of central nodes. However, there is a possibility that central nodes may be distant from consumers, which cannot guarantee a reduction in content retrieval delay.

Additionally, some work employs caching based on a popularity-driven closeness metric (Amadeo et al., 2022) to improve content delays. This approach utilizes a placement selection primarily based on hop distance, resulting in inefficiencies in large network scenarios. In conclusion, as illustrated in Fig. 2, the NDN content delivery network visually represents in-network caching based on existing popularity and freshness metric-driven decision factors.

Figure 2 Existing popularity-aware caching placement.

Problems in existing cache placement strategies

Suboptimal cache placement: Decentralized caching control has led to less-than-optimal cache placement decisions.

One-factor-driven content popularity: Depending on frequency-based content, popularity may not fully capture the nuanced aspects of consumers’ behavior when requesting content.

Popularity and freshness thresholds: Fixed thresholds for content popularity and freshness in caching decisions have resulted in suboptimal cache hit ratios.

Weighted average-based popularity prediction: Inefficient cache miss rates are observed because past popularity records for specific content items are not considered in the weighted moving average used for popularity prediction across different time intervals.

Centrality-metric-based placement selection: Choosing popular caching devices based on betweenness centrality can lead to high delays in content retrieval due to the distance between the source and content consumers.

Hop-distance-based cache placement: Making cache placement decisions based on limited topological characteristics, including hop distance, has resulted in suboptimal content retrieval delays.

Increased packet exchange in NDN: The exchange of traffic within caching devices has increased due to adding a “Flag_bit” in the DATA packet field, caching content near the producer but far from the consumers.

Objectives of the study

In summary, the objectives of this work aim to improve caching placement strategies: Programmable caching devices: Transform caching devices into storage units by implementing a centralized controller-driven caching system to manage caching within network devices.

Multi-factor content popularity: Develop a caching approach that considers factors beyond content frequency, including historical popularity records and request counts, to predict content popularity more accurately.

Time-series model for popularity prediction: Utilize satistical regression-based time series models to predict content popularity within discrete time intervals, thereby reducing cache miss rates by analyzing historical popularity trends.

Topological-features-driven placement: Optimize the placement of popular content by considering topological characteristics like ‘Distance cost,’ ‘Delivery time,’ and ‘Content status.’

Controlled traffic for NDN packets: Implement controlled traffic management using a customized DATA packet with a ‘Cacher’ field, allowing authorized sources to cache the packet while preventing intermediate nodes from caching it.

Evaluating content freshness or validity: Calculate local freshness periods to reduce cache miss rates by piggybacking content onto requested sources.

Research contributions

This research article introduces a framework called ‘PaCPn,’ which utilizes an SDN controller to make informed caching decisions for NDN caching devices. This centralized caching system enhances network efficiency by dynamically managing caching resources and improving the transmission of NDN packets. Unlike complex decentralized caching decision strategies used in previous studies, this centralized approach offers a comprehensive solution that allows the network to quickly adapt to the requested content, optimizing resource utilization and reducing data redundancy. The research article addresses the following research questions: 1) How can the SDN controller be equipped with intelligence to periodically identify a set of popular contents for a content delivery network? • Highly popular content is cached on edge devices using a multi-variant time series model (VAR) embedded within the SDN controller.

• The VAR model predicts popular content based on historical data, including ‘Frequency rates’ and ‘Popularity rates.’

2) What mechanisms does the SDN-controller employ to determine the set of popular nodes for caching the popular contents? • PaCPn employs a closeness-aware heuristic algorithm guided by the SDN controller to place popular content strategically.

• Placement decisions consider factors like ‘distance cost’ (cache node proximity to consumers) and ‘delivery cost’ (time required to deliver content to designated consumers).

3) What strategy does the SDN caching network use to store packets with a valid residual freshness period when the DATA packet is piggybacked to the requested source? • PaCPn improves traffic management by adding fields like ‘Cacher name’ to prevent data duplication and control traffic within caching devices during piggybacked content delivery.

4) What mechanism does the SDN system use to prevent intermediate nodes from caching a copy of content when communication occurs between the requested source and content producer? • Caching devices utilize ‘Generation time’ and ‘Expected time’ fields to calculate each DATA packet’s local residual freshness period, ensuring content validity and freshness.

5) What replacement strategy is employed in this article? • In cases of full caching space, less popular content is replaced with higher-priority content based on popularity values.

• This optimization improves cache hit ratios and reduces content delivery delays.

Proposed solution

The proposed solution, PaCPn, implements on-path caching. When consumers send INTEREST packets, they include a forwarding hint field to determine the optimal path for retrieving the corresponding DATA packet, following the same route for delivery to the requesting consumer. This approach, known as ‘on-path’ caching, is a central focus of this research. It aims to predict the future popularity of incoming INTEREST packets at various time intervals, considering the dynamic evolution of content popularity. To achieve this, a multi-variant vector autoregression (VAR) model estimates the popularity of incoming INTEREST packets at different temporal intervals. Additionally, a heuristic algorithm identifies popular caching nodes for storing the predicted popular content. The network architecture, illustrated in Fig. 3, clearly separates the control and network components. The network architecture is structured into two key layers: a data layer comprising multiple NDN programmable nodes responsible for content caching based on popularity values and a control layer managed by HyperFlow, an SDN controller. Both caching devices and content producers establish out-of-band communication with the HyperFlow controller. This separation enables centralized control and intelligent caching decisions. The HyperFlow controller plays a pivotal role in the network by recording incoming ‘Name-prefix’ at discrete time intervals. It tracks historical values of ‘Request rate’ and ‘Popularity value’ for each requested name component. A VAR popularity model then utilizes these values to enhance content popularity predictions.

Figure 3 Popularity-aware caching in popular programmable NDN nodes (PaCPn).

Minimizing delays is crucial to optimizing content placement within the NDN network and enhancing user experience. This optimization involves considering delivery times between caching points and consumers. A strategic placement approach is developed, factoring in the distance of caching points from consumers, delivery cost, and the status of requested content at evaluating caching points. This strategy establishes a closeness metric guiding the optimal placement of popular content. The SDN controller communicates forwarding signals to content producers, ensuring that popular contents, as predicted by the VAR popularity model, are directed to their closest optimal caching points. This process lies at the core of the proposed framework ‘PaCPn,’ designed for enhanced content delivery. Here are the key points explaining the proposed framework (PaCPn): Data layer: It comprises hierarchical edge nodes that cache popular content, including Ingress nodes for incoming requests, Egress nodes for forwarding requests to producers, and External nodes to fetch producer data. These nodes maintain tables like the ‘CS table,’ ‘PIT table,’ and ‘FIB table.’

Control layer: The controller systematically records content requests in a ‘Content Information Base’ (CIB) table, updating the ‘Request Count’ in real-time and monitoring incoming requests. Additionally, it manages a ‘Historical Table’ (HT), documenting past values of ‘Request count’ and ‘Popularity’ for each content. With its dual functionality, the controller efficiently manages current requests while also analyzing and comprehending historical trends and popularity dynamics associated with various content across the network.

Application layer: Network applications on this layer monitor changes in network topology. The controller employs Link-State routing protocols and the Dijkstra algorithm to create routing tables containing the best paths, which are subsequently installed in the FIB of programmable nodes.

Producer: The content producer caches DATA packets, maintaining a Content Table (CT) with attributes such as ‘Generation_time,’ ‘Version,’ and ‘Intervention_time.’ ‘Generation_time’ signifies the timestamp of DATA packet creation, ‘Version’ indicates the count of updates to the content, and ‘Intervention_time’ represents the duration of data freshness during which a copy remains valid for caching in content routers.

Decision parameters for content caching

The consumer’s requested content is strategically cached on edge devices based on specific decision parameters.

Content popularity

Popularity is one of the most important decision factors to optimize retrieval delays and reduce inter-exchange traffic. However, a time series-based multivariate VAR regression model is considered to predict the popularity of the content delivery network driven by the SDN controller.

(i) Vector autoregression model (VAR): The VAR model-based content popularity prediction has transformed content distribution and network management. VAR models, adept at analyzing historical usage patterns, user preferences, and network conditions, optimize content delivery, resource allocation, and placement (Hyndman & Athanasopoulos, 2018). Proficient in handling time-series multivariable data, VAR models offer valuable insights into content demand patterns over time. They are interpretable, providing insights into relationships between requested contents and historical popularity trends and addressing uncertainty in predicting fluctuating content popularity in NDN environments. As a multivariate time series model extending univariate autoregressive models, VAR predicts requested content popularity based on lagged popularity and recent count variables (Prabhakaran, 2019).

The VAR model stands out for predicting popularity based on multiple variables and their lagged orders for several reasons. In contrast to autoregressive models focusing on singular variables, VAR excels by considering intricate relationships among multiple factors simultaneously. ARIMA, suitable for univariate data, may struggle in complex multivariable scenarios, while LSTM, which demands non-linear relationships, can be computationally intensive and data-demanding. VAR strikes a balance, offering interpretability, scalability, and a nuanced understanding of dynamic relationships influencing content popularity over time.

The VAR model is trained on data collected from edge routers, including distinct consumer requests. The time series data of content popularity and frequency count regarding the requested contents are the independent variables to predict future popularity. Below is a brief explanation of the steps in demonstrating a VAR model (Haslbeck, Bringmann & Waldorp, 2021). (a) Data preparation: The article utilized real data collected from edge caching nodes, detailing the request count against each content transmitted over the edge NDN nodes within discrete time intervals. The popularity variable is constructed discretely for each piece of content using an exponential weighted moving average (EWMA) statistical distribution. To prepare the dataset for VAR modeling, we carry out the following data preprocessing steps: • Timestamp alignment: In the first step of data preparation, each request count is accurately aligned with its corresponding data and period.

• Missing data handling: In the second step, missing values are carefully examined. This step is crucial to ensure the completeness of the dataset, excluding any potential bias in the model training process that could arise from incomplete information. Addressing missing values is imperative for maintaining the integrity and reliability of the VAR model, as any gaps in the data could adversely affect its accuracy and predictive capabilities.

• Data formatting: In the third step, the data is organized into a structured layout. Each timestamp is assigned to a row, and each variable used in the analysis is placed into individual columns. This organized format simplifies data handling and facilitates subsequent modeling.

• Data normalization: In the fourth step, normalization is performed. It involves converting content-name strings to numerical integers in the dataset. The normalization process ensures that all variables are adjusted to a consistent scale, a crucial aspect of VAR modeling where each variable is treated equally. Common normalization methods, such as min-max scaling, are employed in this step.

(b) Lag order selection: Selecting the appropriate lag order is a critical step in VAR modeling, influencing the model’s ability to capture temporal data dependencies. This study employed common information criteria, such as Akaike Information Criterion (AIC) and Bayesian Information Criterion (BIC) to quantitatively determine the optimal lag order, balancing model complexity and explanatory power. Lower AIC and BIC values signify better model fit. The selection process entails fitting VAR models with various lag orders and comparing these criteria to identify the one that best represents the data structure. This approach enhances the VAR model’s ability to capture relevant temporal dynamics, ultimately improving the accuracy of content popularity predictions in the NDN caching system, as discussed in Liu et al. (2019). • Akaike information criterion (AIC): is a measure of the model’s quality that considers the number of parameters used in the model. VAR modeling aims to minimize the AIC value by selecting a lag order with an appropriate balance between model fit and complexity. The AIC is calculated as: AIC=2k−2ln(L) – K is the number of estimated parameters in the model.

– Ln(L) is the natural logarithm of the likelihood function, a measure of how well the model fits the data.

• Bayesian Information Criterion (BIC): is another model fit measure for model complexity. Like AIC, VAR modeling aims to minimize the BIC value to find a lag order that balances model fit and complexity. The BIC is calculated as: BIC=k∗ln(n)−2ln(L) – K is the number of estimated parameters.

– In(n) is the natural logarithm of the number of observations.

– In(L) is the natural logarithm of the likelihood function.

(c) Estimation: in VAR modeling is vital in determining the model’s coefficient. The ordinary Least Squares (OLS) technique is selected as the estimation technique (ERIC, 2021) due to its simplicity and the specific nature of VAR modeling. OLS is a widely used technique that minimizes the sum of squared differences between the observed values and the values predicted by the model. In the context of VAR modeling, the coefficients are estimated in a way that best fits the historical data and minimizes the overall prediction error. • Ordinary least squares (OLS): We have two independent variables, such as Y1 and Y2: Y1, which represents the popularity value, and Y2, which represents request-count, with the lag order of 1 to predict the popularity for next interval, and the OLS estimation can be described as follows: Y1(t)=c1+a11∗Y1(t−1)+a12∗Y2(t−1)+e1(t) – Y1(t–1) and Y2(t–1) are the values of independent variables with their respective timestamp.

– Y1(t) is an estimated value of variable.

– a11, a12, and c1 are the co-efficient metrics that are needed to be estimated.

– e1(t) is a difference between the observed and predicted values at time t.

The OLS estimation involves finding the values of a11, a12, and c1 that minimize the sum of squared differences between the observed values and the values predicted by the model for each equation in the system.

(d) Model diagnostic: Model diagnostic in VAR modeling for content popularity prediction in NDN content delivery networks primarily focuses on assessing the stationarity of the time series data. Stationarity is a critical assumption that ensures that the statistical properties of the data remain consistent over time. Augmented Dickey-Fuller (ADF) is applied toconfirm stationarity. If the data is found to be non-stationary, transformation is required to make it suitable for VAR modeling. Detecting and addressing non-stationary is fundamental in building a reliable VAR model for content popularity prediction in NDN environments.

(e) Model evaluation: Model evaluation is crucial in VAR modeling for content popularity prediction in NDN content delivery networks. Evaluation metrics are essential to ensure the model’s effectiveness. Root Mean Square Error (RMSE) as a metric is suitable for this specific problem as it accurately assesses the model’s ability to capture variations in content popularity over time. RMSE is particularly sensitive to prediction errors, ensuring that prediction errors, whether large or small, are penalized, providing a robust evaluation of the model’s performance.

(ii) Popular node selection: Strategically caching popular content at specific points provides various advantages, such as enhanced network throughput, reduced latency, and increased cache hit rates. The proposed heuristic algorithm selects a list of popular caching points for all requested packets using two metrics: closeness towards the consumer and status of the requested content. (a) Node closeness from the consumers: In the context of edge caching and network optimization, the closeness of edge caching devices to network consumers is crucial for delivering popular content quickly and efficiently. This proximity is typically measured using topological metrics: Delivery time and Distance cost. These metrics are: • Delivery time: It represents the time taken for a DATA packet to be transmitted from the caching point, the ‘Name Data Networking caching device,’ to the set of ingress nodes, which serve as the entry points for all incoming request packets. The packet is then piggybacked from the ingress points to the source cache. (1) Ingress nodes: These are the points where INTEREST packets enter the network, directly connected with the network consumer. (2) Source cache: All the participating caching points in the NDN network store highly requested contents close to the network consumers. (3) Piggybacked: Forwarding the DATA packet along the same route the requested INTEREST packet used for transmission.

(1) deliveryTime(j,ck)=linkcosttodeliverck2

• Distance cost: The incorporation of the distance cost metric in cache placement strategies offers several advantages. Firstly, it helps optimize content delivery by considering the actual physical or logical distance, leading to more efficient and faster data packet transmission within the content delivery network. Secondly, factoring in the hop count between the ingress and caching points enables a more granular and location-aware approach to caching, improving overall content retrieval. Lastly, this metric considers the participating ingress nodes, enhancing the adaptability of the caching strategy to the network’s specific structure and demands.

(2) distanceCost(j,i)=Hopdistancefromjtoinumberofingressnodes

The closeness C of the node j for content ck at time T is denoted as a ratio between Eqs. (1) and (2).

(3) C[T](j,ck)=deliveryTime(j,ck)distanceCost(j,i)

(b) Status of content: This study prioritizes caching popular content at their respective popular nodes by considering the node’s closeness metric and the request count for a specific content. The combination of request count and closeness metric allows us to make more informed decisions by considering the content’s overall popularity and proximity to consumers. This combined approach helps ensure the selected popular nodes are frequently requested and strategically located for efficient content delivery. When both the closeness metric and request count for content ck are high at node j, it is more likely to select node j as the popular node for ck. The content’s status at node j depends on the number of times content ck appears. The content ‘ck’ cache status at node j within the time window T is represented as:

(4) C[T](j,ck)=deliverytime(j,ck)distancecost(j,i)∗Occurrencej(ck)

(iii) Content freshness: When a fresh copy of content arrives at the server or producer, it updates the content’s information. This process involves setting the generation_ time g(t) to the current time, incrementing the version T(v) to indicate it is an updated version, and calculating the intervention_time T(vi) as the difference between the new and previous generation_times (Feng et al., 2022). The generation time reflects the current creation time, the version is incremented, and intervention time quantifies the time difference between the new and previous versions. Additionally, the producer calculates the expected residual time of the content upon the arrival of the updated copy, denoted as E[T(res)]. This calculation considers the difference between the content’s generation_time g(t) and the current time c(t), as well as the ratio of intervention_time T(vi) to content version T(v). E[T(res)]=Tg(t)−T(vi)T(v)−Tc(t). When an INTEREST packet arrives at the content producer, the producer attaches the expected residual_time e(t) and generation_time g(t) to the corresponding fields of the DATA packet. Upon the arrival of the DATA packet at on-path nodes, the node first calculates the residual local caching time presented as L(ck)[T(res)]; the local residual time of content ck.

(5) L(ck)[T(res)]=Tg(t)−Te(t)−Tc(t)

Proposed algorithms

The training process of the VAR popularity model, as outlined in Algorithm 1, follows a set of steps. In the first step (STEP-1), the algorithm initializes the coefficient matrix, representing the weights associated with the timestamp values of each variable. Iterative loops set the initial weights for the VAR coefficients, preparing the model for subsequent estimation.

Algorithm 1 Training the VAR model.

1: Input Parameters:	
2: Data: a matrix where each column represents a time series variable.	
3: Lag: define the Lag order.	
4: num: represents the number of variables.	
5: obs: represents the number of observations. STEP-1	
6: for i to num do	
7:  for j to Lag do	
8:   Coefficient-matrix[i][j] = initial-weights	
9:  end for	
10: end for	
   STEP-2	
11: for i to num do	
12:  for j to Lag do	
13:   X = construct Lag_matrix(data[i], Lag)	
14:   Y = data[i][Lag+1 : obs]	
15:  end for	
16:  Coefficient-matrix[i][j] = solve-coefficient (X, Y)	
17: end for	
   STEP-3	
18: for i to num do	
19:  for t = obs+1 to obs+Forcaststep do	
20:   Popularity[i][t] = PredictPopularity (Coefficient-matrix[i],data[i][t-lag : t-1]	
21:  end for	
22: end for	
   STEP-4	
23: for i to num do	
24:  Calculate RMSE for data[i][obs+1:obs +Forcaststep] and Popularity[i]	
25: end for	

In (STEP-2), the algorithm utilizes the ordinary least squares (OLS) technique to estimate the coefficients. It constructs lag matrices and employs OLS to update the coefficient matrix iteratively. This process educates the model about the relationships between past observations and future values.

In (STEP-3), the algorithm forecasts future content popularity using the trained VAR model. It employs the estimated coefficient matrix to predict popularity for the upcoming time steps. Iterative loops traverse forecast steps, leveraging the model’s understanding of temporal dependencies.

Finally, in (STEP-4), the algorithm evaluates the model’s efficiency by computing each variable’s root mean squared error (RMSE). RMSE assesses the alignment of predicted popularity values with actual values for the forecasted time steps, providing insights into the model’s accuracy and predictive capabilities.

Once the VAR model is trained, it is deployed for real-time predictions in a programmable content delivery network environment. Real-time predictions are generated by loading a pre-trained VAR model with coefficients and hyper-parameters. Data for the current timestamp, including relevant variables and past observations from the HT, is collected. Subsequently, predictions are made using the loaded coefficients matrix to estimate content popularity for the future timestamp.

The closeness-aware heuristic algorithm (Algorithm 2) optimizes content caching in a NDN environment. This algorithm takes as input the set of requested content C r and the set of network cachers N c. It iterates through each requested content and computes a closeness metric CT[Nc[I],Cr[j]] for each caching device in the network. The closeness metric is calculated based on Eq. (4), resulting in a list for each requested content.

Algorithm 2 Closeness_aware heuristic algorithm.

1: Set of requested content Cr	
2: Set of network cachers Nc	
3: Nn←Count(Nc)	
4: Nr←Count(Cr)	
5: for all j to Nr do	
6:  for all i to Nn do	
7:   Compute CT[Nc[i],Cr[j]]−−−−−−eq(4)Eq. (4)	
8:   insert ( Nc[i],C[Nc[i],C_r[j]] ) into the closeness metric named as List[j].	
9:  end for	
10: end for	
11: for all j to Nr do	
12:  for all i to Nn do	
13:   sort List[i] in ascending order using the insertion sort algorithm.	
14:  end for	
15: end for	
16: for all j to Nr do	
17:  for all i to Nn do	
18:   if First item in List[j] = =Nn[i] then	
19:    insert [ Nc[i],Cr[j],CT[Nn[i],Cr[j]] into the Popular cacher List (PL)	
20:   end if	
21:  end for	
22: end for	
23: Repeat	
24: for all j to Nn do	
25:  if First item in PL[j] == Nc[j] then	
26:   insert Nc[j] into (CIB) table cacher field.	
27:   remove content Cr[j] from PL List	
28:  end if	
29: end for	
30: until PL[j] =! 0	
31: Repeat	
32: sending name-prefix with their corresponding cacher name to the content producer.	
33: until (Nr!==0)	

The algorithm then sorts these lists in ascending order using the insertion sort algorithm, effectively arranging caching devices regarding closeness to the requested content. Following this, the algorithm identifies the caching device with the highest closeness value (first item in the list) for each requested content. The chosen caching device and the associated content are inserted into the Popular Cacher List (PL).

In the subsequent iterations, the algorithm updates the Content Information Base (CIB) table by associating each caching device with its corresponding content, effectively populating the estimated popular nodes or NDN caching devices. This process continues until the Popular Cacher List is empty.

The repeated steps involve sending name prefixes with their corresponding caching device names to the content producer, ensuring that the requested contents are directed to their estimated popular caching devices. This iterative process continues until all requested contents are processed.

In summary, the closeness-aware heuristic algorithm efficiently selects the most suitable caching device for each requested content, optimizing the distribution of content across the NDN caching network. The algorithm’s reliance on closeness metrics enhances the effectiveness of content placement, contributing to improved cache hit rates and reduced content retrieval delays.

Processing of INTEREST packet at programmable NDN node

The INTEREST packet processing algorithm (Algorithm 3) in NDN involves crucial steps for handling incoming requests. When an INTEREST packet arrives, the algorithm conducts a lookup in the content store (CS) table. If a match occurs (CS hit), indicating that the requested data is cached, the algorithm searches for the corresponding DATA packet and promptly returns it to the intended interface.

Algorithm 3 INTEREST packet processing.

1: Whenever the INTEREST packet arrived	
2: Lookup into CS table	
3: if (CS hit) then	
4:  search the DATA	
5:  return to the intended interface	
6: else if (PIT hit) then	
7:  adds the incoming interface (from which requests have arrived) in PIT	
8:  wait for DATA packet	
  // do not forward the packet	
9: else if (FIB hit) then	
10:  create PIT entry for requested INTEREST packet	
11:  forward packet according to Flow rule	
12: else	
13:  send OF_Packet_IN message to controller	
14:  install flow rule into FIB by the controller through OF_FLOW_MOD	
15:  forwards the packet to the next station according to FIB	
16: end if	

In scenarios with no match in the CS, the algorithm checks the pending interest table (PIT) to determine if an entry already exists for the requested content (PIT hit). Upon a PIT hit, the algorithm updates the existing PIT entry by adding the incoming interface and awaits the corresponding DATA packet. Notably, the packet is not forwarded in this scenario, optimizing resource utilization.

Subsequently, if there is neither a match in the CS nor a PIT hit, the algorithm examines the forwarding information base (FIB) table. If a match is found in the FIB, the algorithm creates a new PIT entry for the requested INTEREST packet. Following this, the packet is forwarded based on the flow rule specified in the FIB.

When none of the CS, PIT, and FIB conditions are met, indicating a miss, the algorithm triggers an OF_Packet_IN message to the controller. The controller responds by installing a flow rule into the FIB using OF_FLOW_MOD, enabling the algorithm to forward the packet to the next station according to the FIB.

The proposed solution ensures effective request handling through CS, PIT, and FIB tables to enhance the efficiency of INTEREST packet management. It minimizes network latency by dynamically adapting to the current state. Integrating flow rules optimizes content retrieval or packet forwarding based on real-time conditions.

Figure 4 presents efficient INTEREST packet management utilizing CS, PIT, and FIB tables. It ensures effective request handling by retrieving cached content or forwarding packets based on the network’s current state and established flow rules, thereby contributing to a responsive and agile content delivery system.

Figure 4 WorkFlow of INTEREST packet in PaCPn framework.

Processing of DATA packet at programmable NDN node

Algorithm 4 outlines the processing of data packets in a NDN environment. Upon the arrival of a DATA packet, the algorithm performs a Pending Interest Table (PIT) lookup to determine the caching and forwarding decisions. If a PIT hit occurs, indicating an existing entry for the content in the PIT, the algorithm calculates the content’s freshness using Eq. (5).

Algorithm 4 DATA packet processing.

1: Whenever the DATA packet arrived for the PIT Lookup	
2: if (PIT hit) then	
3:  calculate Freshness L(ck)[Tres] – – – – – Eq. (5)	
4: end if	
5: if (cn=pcn and L(ck)[Tres] is non-transient then	
6:  cache DATA packet	
7: else if (cn=pcn and L(ck)[Tres] is transient) then	
8:  fetch the new updated copy of content ck from producer	
9:  forward to intended interface	
10: else if (cn!=pcn and L(ck)[Tres] is transient or non-transient) then	
11:  forward the DATA packet to the next station	
12: else	
13:  drop the Data Packet	
14: end if	

The subsequent steps of the algorithm involve evaluating conditions based on the caching node’s name (c n) and the potential cacher name (p cn), along with the local freshness period L(ck)[Tres] within the NDN environment.

When the names of the caching node and the potential cacher match, and the freshness period is non-transient (indicating an extended period), the algorithm opts to cache the received DATA packet. This decision signifies that the content is fresh and can be stored within the cache for future requests. It indicates recognition by the caching node that it already possesses a valid copy of the content, which is expected to remain valid for an extended period.

In scenarios where the names match, but the freshness period is transient (indicating a short period), the algorithm takes a different course of action. Instead of caching the existing content, it fetches the new updated copy of content ck directly from the producer. Subsequently, it forwards this updated content to the intended interface.

When the names of the caching node and the potential cacher do not match, and the freshness period is either transient or non-transient, the algorithm decides to forward the DATA packet to the next station. This circumstance suggests that the content is not intended for the current caching node but is designated for another destination within the network.

If none of these specific conditions are met, the algorithm concludes by dropping the DATA packet. It implies that the DATA packet does not align with the criteria for caching or forwarding and is not considered further within the processing pipeline. These nuanced decisions are fundamental in managing the storage and dissemination of content within the context of named data networking.

In summary, Fig. 5 effectively manages the caching and forwarding of DATA packets based on specific conditions related to content freshness, producer-consumer relationships, and the local residual time, ensuring optimal utilization of network resources.

Figure 5 WorkFlow of DATA packet in PaCPn framework.

Results and discussion

This section comprehensively evaluates the performance of the proposed caching scheme named PaCPn. The caching scheme designed in this article is compared and analyzed against benchmark schemes using various performance metrics. Our research work has been implemented in the ‘ndnSIM 2.8’ simulation software, presented in Mastorakis, Afanasyev & Zhang (2017). This article follows the experimental topology illustrated in Amadeo et al. (2022) to represent our edge domain, which simulates the flat three-level-based tree topology commonly used in today’s networks. This topology consists of a set of ingress nodes, intermediate nodes, and the egress node that connects the edge domain to the content producer, known as the server. The edge domain establishes out-of-band communication with the SDN-controller, referred to as the ‘HyperFlow controller’ (Hussain et al., 2022), through implementing a C++ programming technique. The controller is integrated into the caching devices by transforming the NDN switches into OpenFlow switches, achieved by enabling the software package known as ‘OFSwitch version 1.3’ (Unicamp, 2023). This transformation makes the caching devices fully programmable.

As presented in Table 2, 1,000 data packets, each comprising 1,024 bytes, are under consideration. The edge domain’s caching capacity is uniformly distributed among nodes, ranging from 0.2% to 0.5% of the total capacity in kilobytes (KB). Requested content follows the Zipf distribution with a skewness parameter (α) set to 0.9. Notably, the simulation encompasses a variable number of consumers, ranging from 200 to 1,200.

Table 2 Simulation parameters.

Parameters	Values	
Content size	1,000 data packets	
Data packet size	1,024 bytes	
Cache size	varying from 0.2 % to 0.5 %	
No of consumers	varying from 200 to 1,200	
Content popularity	Zipf distribution α = 0.9	

The following literature studies are compared to our proposed solution based on various performance metrics. Popularity-aware closeness caching (PaCC): The caching placement scheme in Amadeo et al. (2022) performed the caching decision based on the popularity and closeness-aware metrics.

Dynamically popular content placement (DPCP): The caching placement scheme in Zha et al. (2022) dynamically calculates the popularity threshold for caching the most-requested content in the centralized nodes.

The following performance metrics are considered: Cache hit ratio: The cache hit ratio, a pivotal performance metric in content-centric networks, measures the proportion of requests fulfilled by edge nodes rather than the content producer. This metric is instrumental in assessing the efficiency of caching strategies, considering factors such as cache size, Zipf (α) distribution, and traffic exchanged within edge infrastructure. Experimental results shed light on the intricate relationship between these variables and their impact on the cache hit ratio, providing valuable insights into the effectiveness of the caching infrastructure.

Content retrieval delay: Content retrieval delay, another critical aspect of network performance, gauges the average time consumers experience when retrieving requested content from providers. Various factors influence this metric, including cache size, the number of consumers, and network throughput. Experimental results intricately capture the complex interplay of these variables and their collective impact on content retrieval delays, offering a comprehensive understanding of the temporal dynamics in content delivery within the network.

Cache hit ratio

Emphasizing the cache hit ratio as the primary metric for evaluating popularity placement optimization is essential because it is a key indicator of system efficiency. A heightened cache hit rate reflects efficient cache utilization, indicating that a significant portion of requested content is readily accessible within the network. This optimization aligns seamlessly with the overarching objective of improving system performance and enhancing user experience. By strategically siting popular content closer to users, the cache hit rate is maximized, resulting in minimized retrieval delays and reduced network congestion. The article employs Eq. (6) to evaluate the cache hit ratio, offering a comprehensive and quantitative measure of the efficacy of cache placement strategies. This performance metric facilitates a thorough assessment, considering factors such as content availability, network responsiveness, and the overall optimization of content delivery in content-centric networks.

(6) CHr=∑i=1N(numi)Requestedcontent

V=Setofnodesinthenetworktopology.

Numi=Numberofhitsatnodei.

Impact of hit ratio based on cache capacity

The cache size significantly influences the cache hit rate. Generally, as the size of the cache increases, the likelihood of storing frequently accessed or popular items also rises. This, in turn, results in an increased cache hit rate. A larger cache enables a more extensive retention of frequently requested content, diminishing the necessity to fetch data from the underlying storage or network. Consequently, this improvement enhances the overall efficiency of the caching system.

According to Fig. 6, performance metrics vary as the cache size of the edge domain ranges from 0.2% to 0.5%. As anticipated, increased cache size enhances performance across all considered schemes. Larger storage capacity enables more content storage, thereby reducing delivery time and hop count, leading to advantages in NDN traffic exchange.

Figure 6 Impact of hit ratio by varying cache size.

Figure 6 illustrates that the DPCP solution exhibits the lowest hit ratio compared to other schemes. It is attributed to the DPCP solution’s focus on calculating content popularity using the request count, which may not effectively improve the hit rate given dynamically changing popularity trends for specific content items. On the other hand, the PaCC caching scheme performs slightly better than DPCP. When distributing popular content among edge node caches, it considers the requested count and popularity at the previous interval. Nevertheless, it fails to capture the history of requesting content items, increasing the cache miss ratio.

In contrast, PaCPn determines the content popularity based on lagged ‘Popularity-score’ and ‘Request-count’ values. The proposed controller-aware VAR popularity model is trained concerning these factors to anticipate future popularity.

Additionally, when the requested content is not cached in the network, it first calculates its local-residual freshness period when receiving it from the producer to limit the miss ratio while contributing to an improved cache-hit ratio. In Fig. 6, ‘DPCP’ achieves a cache hit rate of 63%, while ‘PaCC’ reaches 75% with an expanding edge domain cache. ‘PaCPn’ stands out by achieving a 90% hit rate through dynamic updates of content popularity using a VAR model on an SDN controller.

Impact of hit ratio based on Zipf distribution

We observe the impact of the Zipf distribution parameter α on the cache hit ratio. As α increases, user requests tend to concentrate on the most popular content. This skewed distribution, associated with higher α values, indicates that a small subset of items is frequently requested, creating a long tail’ distribution. Consequently, during a sudden surge in requests for popular content, the cache hit rate tends to rise. This behavior aligns with the Zipf distribution’s representation of skewed popularity, emphasizing the significance of comprehending and adjusting α when optimizing caching strategies to capture and serve the most requested content efficiently.

In Fig. 7, the ‘DPCP’ solution shows the lowest hit rate compared to the others. In contrast, the ‘PaCC’ solution performs slightly better than ’DPCP,’ thanks to the dual-factor-based content popularity prediction used in calculating content popularity. Conversely, the proposed solution, ‘PaCPn,’ achieves the highest hit ratio compared to the others. It accomplishes this by selecting popular content based on its historical record, effectively capturing user attraction towards them. Under the considered settings in Fig. 7, ‘DPCP’ achieves a hit rate of 54%, ‘PaCC’ reaches 68% when the Zipf parameter α increases, but ‘PaCPn’ excels at 80% by considering changing consumer behavior for improved predictions.

Figure 7 Impact of hit ratio by varying Zip( α) distribution.

Impact of hit ratio based on exchanged CR traffic

Network traffic plays a pivotal role in shaping cache hit rates in an NDN environment, where content retrieval is based on names rather than locations. Efficient and streamlined data transfer in optimal traffic conditions increases cache hit rates. When network traffic is well-managed, the popular selection of content (driven by the multivariate VAR model) is readily available in local caches, minimizing latency associated with fetching data from distant sources while enhancing the cache hitting rate.

In the event of suboptimal traffic conditions, such as congestion or inefficient data transfer, there is the potential for a decrease in cache hit rates. It underscores the need for the programmable nature of the NDN environment, which becomes particularly crucial in addressing challenges associated with suboptimal traffic conditions. The proposed solution leverages programmable mechanisms, effectively mitigating congestion and optimizing data transfer. It, in turn, ensures a more robust and responsive caching system even under less-than-ideal traffic conditions.

As a result, it is evident that ‘PaCPn’ excels in comparison to ‘DPCP’ and ‘PaCC’ by considering ‘Request-count’ and ‘Popularity score’ over pre-intervals to populate cache nodes with popular content, increasing the hit ratio and reducing server load. This approach significantly boosts the cache hit ratio, substantially reducing the amount of NDN traffic within the network. As more user requests are efficiently served from the cache due to the higher hit ratio, there is less dependency on external sources, resulting in a reduced exchange of traffic within the NDN network. Figure 8 shows DPCP achieving a 55% hit rate, PaCC at 65%, and PaCPn surpassing both with an 80% hit ratio. This superior performance is attributed to considering previous lag values and enhancing new content popularity prediction.

Figure 8 Impact of hit ratio by varying CR traffic.

Content retrieval delay

Content retrieval delays are critical for evaluating optimal placement in the programmable NDN caching environment. Based on the time taken to retrieve requested content from the cache, these delays significantly impact the efficiency of cache placement. The strategy involves storing frequently requested content in proximity-aware caches, reducing the need for extensive data transmission. It not only enhances the user experience but also optimizes network resource utilization. In the programmable NDN caching landscape, focusing on reducing retrieval delays is key to overall performance, ensuring optimal content delivery responsiveness. The retrieval delays can be computed using the formula defined in Eq. (7).

(7) RD=∑i=1(RDi)|reqc||consumers|

RDi=Retrievaldelayofasinglecontenti.

reqc=Totalamountofrequestedpackets.

sub=Totalnumberofcontent consumers.

Impact on retrieval delays based on cache capacity

The influence of cache size on content retrieval delays is a significant factor in shaping the efficiency of a content delivery system. In essence, the size of the cache directly impacts the system’s ability to store and quickly retrieve frequently requested content. A larger cache facilitates the storage of popular items, reducing the need to fetch data from underlying storage or network sources. Consequently, with a generous cache size, the content retrieval process becomes faster and more responsive as the likelihood of finding requested data in the cache increases. On the contrary, a smaller cache may lead to more frequent instances of content not being readily available, resulting in longer retrieval delays as the system resorts to fetching data from external sources. It underscores the critical role of optimizing cache size for minimizing content retrieval delays and enhancing the overall efficiency of a content delivery system.

In Fig. 9, DPCP shows the highest retrieval delays as the cache size increases, primarily due to its dependence on content popularity score, which may not effectively reduce delays with larger caches. PaCC performs slightly better by considering popularity and distance but is compromised by a hop count-based node selection approach, lacking consideration for factors like link cost and latency, leading to inefficiencies. Multiple updates in PaCC for optimal content placement make it computationally expensive, especially with larger caches.

Figure 9 Impact on retrieval delay by varying cache size.

In contrast, PaCPn employs a centralized caching policy managed by the SDN controller to minimize computational costs and optimize caching decisions, resulting in superior performance with larger cache sizes. The PaCPn caching algorithm makes placement decisions based on distance cost, delivery time, and popularity status to reduce delivery times and network delays. It uses a closeness-aware metric considering ‘content delivery time’ and ‘content distance cost.’ Delivery cost reflects the link cost, and distance cost measures the caching node’s proximity to the user. The strategy also accounts for content popularity, optimizing placement for frequently requested content.

In comparative performance evaluations, the dynamic popular content placement (DPCP) strategy achieves a retrieval delay of 0.07 ms, showcasing its efficiency. The popularity-aware caching (PaCC) strategy attains even lower delays of 0.06 ms, which is particularly noteworthy with an expanding edge cache size. However, the PaCPn algorithm stands out, surpassing all metrics by demonstrating superior performance with delays reduced to an impressive 0.046 ms.

Impact on retrieval delays based on number of consumers

The impact of the number of consumers on content retrieval delays is a crucial factor in determining the responsiveness of a content delivery system. As the number of consumers increases, the system experiences a higher demand for content, potentially leading to increased retrieval delays. A more extensive consumer base results in more frequent requests for various content, placing additional demands on the system’s resources. In scenarios where the optimized placement strategy is not adequately scaled to accommodate the growing number of consumers, retrieval delays may occur as the system struggles to meet the heightened demand. Conversely, retrieval delays can be minimized in an environment with a well-scaled infrastructure capable of handling a more extensive consumer base. Therefore, optimizing the placement strategy to align with the number of consumers is crucial for ensuring efficient content retrieval and maintaining satisfactory performance even under increased demand.

As illustrated in Fig. 10, all the considered schemes exhibit lower retrieval delays as the number of consumers increases. Especially notable is the proposed solution, which demonstrates the lowest delay compared to other schemes when responding to DATA requests from consumers. It underscores the potential for optimizing content retrieval delays by intelligently managing caching decisions and placement, especially in scenarios with increased consumers and concentrated requests for popular content.

Figure 10 Impact on retrieval delay by varying consumers.

The solution ensures that even with the growing number of consumers, the most popular content is cached at the closeness edge points, reducing retrieval delays and enhancing overall content delivery efficiency. The experimental results reveal distinct performance variations among caching policies. DPCP exhibited a retrieval delay of 0.09 ms, PaCC demonstrated efficiency with a content delivery delay of 0.06 ms, and notably, our proposed caching scheme outperformed both, achieving an impressive delivery rate of 0.04 ms. This superior performance is attributed to our innovative heuristic algorithm, dynamically allocating caching resources to popular content. The dynamic allocation ensures strategic placement, reducing retrieval delays and enhancing the content delivery experience.

Impact on retrieval delays based on network throughput

The influence of system throughput on content retrieval delays is of crucial significance, directly impacting the speed and efficiency of delivering requested content. System throughput, representing the number of data packets processed within a given timeframe, is crucial in determining how swiftly content can be retrieved and transmitted to consumers. Higher system throughput correlates with faster content delivery, minimizing retrieval delays and ensuring a more responsive user experience. In contrast, lower system throughput can lead to increased retrieval delays as the system struggles to process and transmit content efficiently. Therefore, optimizing system throughput is instrumental in achieving prompt and reliable content retrieval, directly contributing to users’ overall performance and satisfaction interacting with the content delivery system. As a result, the delivery time is reduced because devices are not individually making cache placements and forwarding decisions; the controller dynamically implements closeness-aware placement decisions at regular intervals, unlike the other schemes in which distributed placement decisions are performed through the betweenness centrality approach. This approach allows a larger content volume to be efficiently delivered within shorter time frames, enhancing the overall network throughput of DATA packets in bits per second (DPBS). Figure 11 shows that the DPCP solution reaches a minimum delay of 0.07 ms. The other scheme achieves a minimum retrieval delay of 0.05 ms as the system throughput increases. On the other hand, the proposed solution PaCPn achieves the lowest content delivery rate of 0.033 ms due to its centralized and dynamically managed placement of popular content in popular NDN devices.

Figure 11 Impact on retrieval delay by varying system throughput.

Conclusions and future work

In-network caching can significantly reduce the traffic overhead for content producers, particularly when retransmissions are required due to packet loss. Placing copies of requested content in the edge domain can significantly reduce network load and transmission latency.

This work introduces a novel caching placement scheme that considers distance, time latency, and content frequency when selecting the popular node or location for caching the most requested content. A regression VAR model predicts this selection. The SDN controller drives the proposed caching placement scheme, which monitors consumer request temporal and regional patterns. It periodically updates the content provider’s caches using the previous lag values of multi-variant attributes in the proposed popularity model. To determine the optimal placement for each requested name prefix, we have introduced a heuristic algorithm driven by closeness metrics. This algorithm primarily considers the ratio of ‘Content delivery cost’ and ‘Content distance cost,’ associated with the ‘Status of content frequency,’ when calculating the closeness metric for each requested INTEREST packet.

To evaluate the performance of our proposed solution, we employed a network topology comprising programmable NDN caching devices. In our experiment, we assessed the effectiveness of the proposed cache placement scheme in terms of content retrieval delay and cache hit ratio. The experimental results showed that our proposed solution, ‘PaCPn,’ improved the cache hit rate by 20% and reduced the retrieval delay by 28% compared to existing solutions. These substantial enhancements underscore the practical impact of our proposed caching scheme on the efficiency of content distribution within the network, emphasizing its potential to elevate user experience and reduce network resource utilization.

In future work, the proposed solution, which focuses on VAR-based popularity prediction populated in the distance and time delivery-aware programmable NDN devices, can be extended by integrating and testing it across diverse domains. Extending our caching solution to these diverse domains holds promising benefits, as it could enhance the adaptability and efficiency of content delivery in scenarios such as IoT communication (Naeem et al., 2018), edge cloud services (Song et al., 2017), and fog computing applications (Amadeo, Campolo & Molinaro, 2016). By addressing specific challenges and intricacies within these contexts, our proposed caching scheme has the potential to contribute significantly to optimizing content distribution across various emerging technologies, paving the way for more robust and responsive network architectures.

Supplemental Information

Supplemental Information 1 Code related to the experiments performed.

Additional Information and Declarations

Competing Interests

Author Contributions

Data Availability

Ivan Miguel Pires is an Academic Editor for PeerJ Computer Science.

Firdous Qaiser conceived and designed the experiments, performed the experiments, analyzed the data, performed the computation work, prepared figures and/or tables, authored or reviewed drafts of the article, and approved the final draft.

Mudassar Hussain conceived and designed the experiments, performed the experiments, analyzed the data, performed the computation work, prepared figures and/or tables, authored or reviewed drafts of the article, and approved the final draft.

Abdul Ahad conceived and designed the experiments, performed the experiments, analyzed the data, performed the computation work, prepared figures and/or tables, authored or reviewed drafts of the article, and approved the final draft.

Ivan Miguel Pires conceived and designed the experiments, performed the experiments, analyzed the data, performed the computation work, prepared figures and/or tables, authored or reviewed drafts of the article, and approved the final draft.

The following information was supplied regarding data availability:

The code is available in the Supplemental File. There was no data used in the article. The results were obtained using the realtime simulation.

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
