# Peer review of "Controller-driven vector autoregression model for predicting content popularity in programmable named data networking devices"

_PeerJ Computer Science, doi:10.7717/peerj-cs.1854_

## Round 0.1 · original submission · Major Revisions

With the reviewers' comments, I have some questions and suggestions. The function of Named Data Networking (NDN) remains unclear. The working process of NDN needs to be rewritten in detail. The scheme also affects the naming system of NDN since it operates with name-based routing. The authors need to explain their scheme from a naming perspective as well.

**Language Note:** The review process has identified that the English language must be improved. PeerJ can provide language editing services - please contact us at copyediting@peerj.com for pricing (be sure to provide your manuscript number and title). Alternatively, you should make your own arrangements to improve the language quality and provide details in your response letter. – PeerJ Staff

Reviewer 1 ·

Basic reporting

The paper titled 'Popularity-aware Caching in Popular Programmable NDN Nodes (PaCPn)' introduces a pioneering approach to optimizing content delivery within Named Data Networking (NDN) infrastructures. Its coverage of related work, problem statement, proposed solution, analysis, and results is comprehensive, greatly aiding in understanding the research problem and its proposed solution. While the manuscript demonstrates good clarity, it would greatly benefit from a thorough review by an English language expert for further refinement. Overall, the paper is recommended for acceptance with minor revisions.

Experimental design

The research design showcases several strengths that contribute to its overall clarity and depth, yet there are areas that could benefit from further elaboration:
1. The problem statement is notably well-articulated and effectively supplemented by an explanatory diagram. This diagram aids in providing a visual understanding of the problem, enhancing its clarity and accessibility.
2. The proposed solution is meticulously presented with a comprehensive set of algorithms and detailed explanations. However, further granularity could be beneficial in outlining the specific intricacies of the proposed solution.
3. The research questions are appropriately defined and relevant. Enumerating these questions would contribute to a clearer structure and facilitate a more precise understanding.
4. The inquiry into the advantages of SDN within NDN environments and its impact on traditional networking environments could benefit from further elaboration.
5. The rationale behind selecting the Vector Autoregression (VAR) model over other machine learning or deep learning models is pivotal. A detailed explanation elucidating the reasons for this choice, along with comparative advantages of VAR in the study's context, would strengthen the methodology.

Validity of the findings

1. The research findings exhibit notable improvements compared to existing schemes, validating the effectiveness of the proposed technique.
2. Enhancing visual representation by including units in legends would improve clarity and bolster the validity of the findings.
3. A rationale for selecting cache hit ratio and Content Retrieval delay would strengthen the validity by substantiating their relevance in assessing the technique's effectiveness.
4. Detailed demonstrations of the technique's impact on cache hit rates and content retrieval delays, alongside its potential optimizations for edge infrastructures, validate the findings and showcase their significance.

Additional comments

No comments

Reviewer 2 ·

Basic reporting

The manuscript, "Controller-driven VAR model for predicting content popularity in programmable NDN devices" introduces an innovative approach to enhance content delivery within Named Data Networking (NDN) infrastructures, particularly advancing NDN content caching and offering potential optimizations for edge infrastructures. However, the manuscript requires Major Revisions, and the authors are advised to address the comments and improve the manuscript.
1. Although well-written, a careful proofreading is advised to enhance English grammar and overall composition.
2. Add more approaches related to this research work in the literature review section.
3. Enhance the quality of diagrams to improve visual clarity.
4. Add a table containing definitions of all terms being used in the paper.

Experimental design

1. Problem Statement needs to be clearly defined. Why is content popularity optimization considered crucial for efficient caching strategies in network environments?
2. The proposed solution requires a more in-depth technical discussion, employing algorithms to elucidate how it implements Popularity-aware Content Caching in Programmable NDN Nodes, especially considering the integration of SDN.

Validity of the findings

1. While the experimental results demonstrate substantial improvements over existing techniques, it's essential to elucidate the rationale behind selecting cache hit ratio and Content Retrieval delay for result analysis.
2. Consider including units in legends for better comprehension within the results section.

Additional comments

1. Expand on future work to offer more comprehensive insights.

---

## Round 0.2 · accepted · Accept

The paper is now acceptable.

Reviewer 1 ·

Basic reporting

The authors have updated the manuscript according to the comments.

Experimental design

The authors have updated the manuscript according to the comments.

Validity of the findings

The authors have updated the manuscript according to the comments.

Reviewer 2 ·

Basic reporting

The manuscript is comprehensively revised as per comments and require no further revisions. The paper is recommended for publication.

Experimental design

The experimental design is improved as per comments and requires no further revisions. The paper is recommended for publication.

Validity of the findings

The findings of study are improved as per comments and manuscript is comprehensively revised. The paper is recommended for publication.

Additional comments

No additional comments.